# The Interplay between Bioactive Sphingolipids in the Psoriatic Skin and the Severity of the Disease

**DOI:** 10.3390/ijms241411336

**Published:** 2023-07-12

**Authors:** Mateusz Matwiejuk, Hanna Mysliwiec, Bartłomiej Lukaszuk, Marta Lewoc, Hend Malla, Piotr Mysliwiec, Jacek Dadan, Adrian Chabowski, Iwona Flisiak

**Affiliations:** 1Department of Dermatology and Venereology, Medical University of Bialystok, 15-540 Bialystok, Poland; maaateusz1996@gmail.com (M.M.);; 2Department of Physiology, Medical University of Bialystok, 15-222 Bialystok, Poland; 31st Clinical Department of General and Endocrine Surgery, Medical University of Bialystok, 15-276 Bialystok, Poland

**Keywords:** psoriasis, ceramide, sphingosine-1-phosphate, sphinganine-1-phosphate, sphinganine, sphingosine

## Abstract

Psoriasis is a complex chronic immunologically mediated disease that may involve skin, nails, and joints. It is characterized by hyperproliferation, deregulated differentiation, and impaired apoptosis of keratinocytes. Sphingolipids, namely ceramide, sphingosine-1-phosphate, sphingosine, sphingomyelin, and sphinganine-1-phosphate, are signal molecules that may regulate cell growth, immune reactions, and apoptosis. Fifteen patients with psoriasis and seventeen healthy persons were enrolled in the study. Skin samples were taken from psoriatic lesions and non-lesional areas. Tissue concentration of ceramides, sphingosine-1-phosphate, sphingosine, sphingomyelin, and sphinganine-1-phosphate was measured by liquid chromatography. We assessed that all levels of ceramides, sphingosine-1-phosphate, sphingosine, sphingomyelin, and sphinganine-1-phosphate were higher in lesioned psoriatic skin than in non-affected skin. The profile of bioactive lipids in the lesional skin of patients with psoriasis differed significantly from non-involved psoriatic skin and skin in healthy subjects.

## 1. Introduction

Psoriasis is a chronic inflammatory immune-mediated disease that is widespread around the whole world [1]. Its global prevalence is between 0.27% and 11.4% in adults and it appears more frequently in adults than in children [2]. Psoriasis may be divided into a few significant subgroups, e.g., erythrodermic, guttate, pustular, inverse, and plaque (the most common (about 90% of cases)) [3]. The most classic symptoms of plaque psoriasis are erythematous well-demarcated lesions that are generally covered by silvery scales. Those plaques may appear on large surfaces of the patient’s skin and may merge together. Furthermore, typical psoriatic locations may include the extensor surfaces of the limbs, the scalp, and the lumbosacral region [4]. In typical psoriatic skin lesions, deregulated differentiation of epidermal keratinocytes (parakeratosis and acanthosis) and infiltration of immune cells into the deeper layers of the skin may be observed [5]. Interestingly, a decreased apoptosis of keratinocytes in psoriatic lesions has been noted [6]. Furthermore, keratinocytes derived from psoriatic plaques were resistant to apoptosis in comparison with keratinocytes in healthy skin [7].

Psoriasis may not only affect the skin but also joints and nails and may co-occur with various systemic conditions [8].

Mainly, due to psoriatic pro-inflammatory etiology, it may coexist with different components of metabolic syndrome, for example, hyperlipidemia, hypertension, obesity, and insulin resistance. Over time, these conditions may lead to the development of type 2 diabetes, atherosclerotic disease, coronary artery disease, or myocardial infarction [9].

Psoriatic patients are commonly diagnosed with abnormal lipids levels. Serum amounts of triglycerides, total cholesterol, and very low-density lipoprotein (VLDL) cholesterol and low-density lipoprotein (LDL) cholesterol are raised in comparison with healthy people. However, high-density lipoprotein (HDL) cholesterol is substantially decreased in patients suffering from psoriasis [10].

Until now, the molecular mechanisms underlying psoriasis, its progression, and concomitant diseases are still unclear despite many recent studies. Sphingolipids are one of the major groups of eukaryotic lipids [11]. Sphingolipids are lipids that contain a sphingoid core. The sphingoid basis is produced with the connection of fatty acids (mainly palmitate) and amino acids (principally serine) [12]. Sphingolipids are an essential subgroup of the lipid mediator family, with both signaling and structural capabilities [13]. The abundance of sphingolipid tasks is broad and applies to the majority of the main features of cell biology, for instance, roles in cell growth, the process of cell death, the cell cycle, immune activity, nutrient uptake, cell adhesion, inflammation, metabolism, angiogenesis, responses to multiply stressors and autophagy, and reactive oxygen stress stimuli. Various types of sphingolipids have been defined, for example, ceramide (CER), sphingosine-1- phosphate (S1P), sphingosine (SFO), ceramide-1-phosphate (C1P), sphingomyelin, galactosylceramide, glucosylceramide, and lactosylceramide [11]. The most active sphingolipids are sphingosine-1-phosphate and ceramide, which are known for various signaling roles. Ceramides are more involved in inflammation, stress responses, cell cycle arrest, apoptosis, and necrosis. Nevertheless, sphingosine-1-phosphate is a signaling molecule that plays a role in cell growth, proliferation, differentiation, and migration. Moreover, the binding of sphingosine-1-phosphate to its cell surface receptors initiates angiogenesis [14]. Reduced levels of ceramide have been linked with different skin diseases involving barrier disruption and dryness, such as xerosis, atopic dermatitis, and psoriasis [15]. Elevated levels of the most analyzed ceramides (C_16:0_, C_18:0_, C_20:0_, C_22:0_, and C_24:1_) have been observed in patients suffering from psoriasis (in the skin and plasma) compared with healthy controls. Furthermore, a decreased level of C_12:0_-sphingomyelin has been spotted in severe psoriatic patients in comparison with healthy controls. Interestingly, the C_12:0_-ceramide was the only lipid molecule that was lowered in severe psoriasis patients compared with healthy controls There were no differences observed in the levels of the hexosylceramide or lactosylceramide. Levels of sphingomyelin were impaired in psoriatic skin in a fatty-acid chain length-dependent manner, with the growth of C_16:0_-, C_24:1_-, and C_24:0_-sphingomyelins. Levels of C_12:0_-sphingomyelin were lower in non-lesional skin vs. lesional and control skin [13].

Interestingly, serum sphingolipids (mostly ceramides) may play a role as markers of atherosclerotic cardiovascular disorders and metabolic diseases. Plasma levels of C26 and C24 ceramides and deoxy-C24 ceramide have been linked with diabetic neuropathy. Moreover, elevated C18:1 and C18:0 ceramides have also been described as relevant markers of major undesirable cardiovascular effects in healthy people; especially, C18:1 ceramide is known to have a ratio of high levels of necrosis after coronary angiography procedures. C24:1 ceramide and sphingomyelin have been strongly linked with cardiovascular death rates. Elevated amounts of plasma C22:0 and C24:0 ceramides may predict a smaller enhancement in verbal memory in reply to exercise in patients with coronary artery disease where these skills are lowered. Alternatively, the serum sphingosine-1-phosphate level in atherosclerotic disease was inversely correlated, where deoxysphingolipids functioned as a biomarker in diabetes [11].

Additionally, sphingosine-1-phosphate inhibited keratinocyte proliferation and induced migration and differentiation [16]. Moreover, the level of sphingosine was heightened in psoriatic skin (one of the precursors of sphingosine-1-phosphate) [17].

In recent years, studies have mainly focused on analyzing serum lipid levels in patients with psoriasis. However, it is not yet fully understood what lipid abnormalities exist in the skin tissue of individuals affected by this condition. The present study aims to assess the interplay between bioactive sphingolipids in the psoriatic skin and the severity of the disease.

## 2. Results

### 2.1. Study Population

A total of 15 patients (7 males and 8 females) with active plaque-type psoriasis and 17 healthy patients (11 males and 6 females) were included in the study. The median age in the control group was 42, ranging from 23 to 84 years; the median range in the psoriatic group was 51, ranging from 23 to 71 years. The average duration of psoriasis was 24 years. The median body mass was 87.0 (82.0–94.0) (kg). The mean height was 174.0 (162.0–176.0) (cm). The median BMI was 28.74 (27.72–30.35). Most of the patients (*n* = 8) were overweight (53.33%), five (33.33%) suffered from obesity, and two (13.3%) had normal weight. In the examined group, 1 (6.67%) patient had a mild (PASI < 10) form of psoriasis, 10 (66.67%) suffered from moderate psoriasis (PASI 10–20), and 4 (26.67%) had a severe (PASI > 20) form of psoriasis. Table 1 summarizes the main clinical features of the psoriatic group and the control group.

### 2.2. Sphingolipid Parameters

The median concentration of sphingosine found in the psoriatic lesional skin of patients (1.72 pmol/mg) was found to be statistically and significantly higher (*p* < 0.05) than both the non-lesional skin of psoriasis patients (0.38 pmol/mg) and the skin of healthy individuals (0.27 pmol/mg). Likewise, the median ceramide concentration in the affected lesional skin (68.4 pmol/mg) was also significantly higher (*p* < 0.05) than the non-lesional skin (16.08 pmol/mg) and skin of healthy subjects (6.58 pmol/mg). Additionally, the concentration of other parameters determined in this study, such as sphinganine, sphingosine-1-phosphate, and sphinganine-1-phosphate, were found to be significantly higher in lesional skin and showed a slight non-significant increase in non-lesional skin compared with the skin of healthy individuals. Figure 1 provides a summary of the primary clinical features observed in psoriatic patients, including lesional and non-lesional skin samples, as well as the control group.

Furthermore, as illustrated in Figure 2, we observed a positive Pearson’s correlation between psoriatic skin induration and sphinganine, as well as between induration and sphinganine-1-phosphate, both of which were statistically significant with *p*-values of less than 0.035 and 0.046, respectively. Moreover, we identified a positive Pearson’s correlation in non-lesional skin of psoriatic patients between induration and sphingosine-1-phosphate and induration and sphinganine-1-phosphate.

Additionally, we observed statistically significant positive correlations between psoriasis area and severity index_total; sphingosine, sphinganine, and sphinganine-1-phosphate; as well as between induration of skin lesions and sphinganine and induration and sphinganine-1-phosphate in lesional skin samples of psoriatic patients.

## 3. Discussion

We studied sphingolipid levels in psoriatic patients concerning clinical and laboratory data.

In this study, we evaluated and compared the concentrations of bioactive sphingolipids in both lesioned and non-lesioned skin samples from psoriatic patients, which, to the best of our knowledge, has not been previously explored. Our findings revealed that levels of sphingosine, sphinganine, sphingosine-1-phosphate, sphinganine-1-phosphate, and ceramide were significantly higher in the lesional psoriatic skin samples compared with the non-affected skin samples in patients with psoriasis.

Our study identified a statistically significant positive correlation between induration and both sphinganine and sphinganine-1-phosphate in non-lesioned skin samples of psoriatic patients.

Our study demonstrated an interesting finding that the SFA/SFO ratio was highest in the control group, lower in the non-lesioned skin of psoriatic patients, and the lowest in the lesioned skin of psoriatic patients.

### 3.1. The Role of Sphingosine-1-Phosphate

In our study, we found that sphingosine-1-phosphate levels were elevated in skin samples obtained from patients with psoriasis compared with individuals with healthy skin from patients with hernia inguinal. These findings were consistent with previous studies that also reported increased sphingosine-1-phosphate levels in the skin and serum of psoriasis patients. However, our study revealed a novel finding, indicating that the sphingosine-1-phosphate concentration was elevated specifically in skin lesions. In non-lesioned skin, sphingosine-1-phosphate levels were significantly lower than in psoriatic lesions, although still comparable to the levels found in healthy skin.

Sphingosine-1-phosphate is a signaling lipid known for its crucial role in regulating inflammation, angiogenesis, and vascular permeability [18]. In our study, we discovered a positive correlation between sphingosine-1-phosphate and induration in psoriatic skin lesions, which could be attributed to its involvement in promoting inflammation and the angiogenesis process. Interestingly, elevated plasma levels of sphingosine-1-phosphate have been observed not only in psoriasis but also in obesity when compared with healthy controls. This suggests that sphingosine-1-phosphate may have broader implications in various pathological conditions beyond psoriasis. Indeed, sphingosine-1-phosphate has been found to correlate with metabolic irregularities such as insulin resistance and adiposity. In line with this, the authors of the study identified a significant association between sphingosine-1-phosphate and features of metabolic syndrome, including body fat percentage, waist circumference, total, and LDL cholesterol levels, and fasting plasma insulin [19]. Additionally, it was observed that psoriatic patients had elevated serum concentrations of sphingosine-1-phosphate along with higher serum alanine aminotransferase (ALT) levels compared with healthy controls [20]. These findings suggest a potential link between sphingosine-1-phosphate, metabolic disturbances, and liver function in psoriatic patients. Furthermore, a statistically significant positive correlation was identified between the severity of psoriasis and the serum levels of sphingosine-1- phosphate both before and after treatment with narrow-band ultraviolet B (NBUVB) therapy (r = 0.374, *p* = 0.003). However, no significant correlation was found between disease duration or patient age and sphingosine-1-phosphate levels (r = 0.393, *p* = 0.765) [21]. This suggests that sphingosine-1-phosphate serum levels may serve as a potential marker for assessing the severity of psoriasis and monitoring the effectiveness of NBUVB treatment.

Sphingosine-1-phosphate is well-established for its ability to induce keratinocyte differentiation, exhibit anti-proliferative and pro-inflammatory effects, and inhibit epidermal cell growth in mouse models of psoriasis, as described by Vaclavkova et al. [22]. Ponesimod, on the other hand, is an orally administered selective modulator of sphingosine 1-phosphate receptor 1 (S1PR1). It functions by blocking the outflux of T cells from lymphoid organs. Based on our findings and the results reported by other researchers, targeting S1P1 to inhibit the gathering of pathogenic lymphocytes in the skin and circulation appears to be a promising therapeutic approach for the future treatment of psoriasis. This suggests that modulating the sphingosine-1-phosphate pathway could potentially offer new avenues for managing and improving psoriasis treatment outcomes. Interestingly, in the phase 2 clinical trial, by week 16, the PASI75 response to ponesimod was 46.0% (with a dose of 20 mg) and 48.1% (with a dose of 40 mg) compared with a 13.4% response for the placebo. By week 28, PASI75 was 71.4% (20 mg) and 77.4% (40 mg), respectively [22]. D’Ambrosio et al. [23] reported that ponesimod presented encouraging outcomes in phase II studies in relapsing–remitting psoriasis and multiple sclerosis. Their study suggested that lymphocyte depletion may form the basis of action for a selective S1P1 receptor modulator in chronic inflammatory and multiple autoimmune impairments for instance psoriasis. The amount of peripheral blood T cells and B blood cells decreased following the intake of 8 mg of ponesimod. CD3+ and CD 20+ counts were decreased significantly following the administration of 20–75 mg of ponesimod. This study presented that ponesimod may decrease some of the inflammatory cells, which may reduce the inflammation process in psoriasis, which is one of the backgrounds of this skin disease [23].

Another studied sphingosine-1-phosphate receptor modulator is fingolimod. De Biase et al. [24] showed a case where fingolimod led to a remission of psoriatic skin lesions in a 27-year-old patient with multiple sclerosis. Interestingly, psoriasis and multiple sclerosis are both linked with Th1 and Th17 cells. In previous studies, fingolimod has been shown to reduce the levels of sphingosine-1-phosphate, inhibit the exocytosis of lymphocytes from lymphoid tissues, and decrease the number of Th17 lymphocytes in peripheral blood, which may play a role in the treatment of multiple sclerosis and psoriasis. As a consequence, fingolimod might be a therapeutic solution if these two diseases coexist [24]. Okura et al. [25] showed that fingolimod was efficacious in improving imiquimod-induced psoriasiform dermatitis in mice, both clinically and histologically. Moreover, the amount of IL-17A was depleted in mice skin treated with fingolimod than in mice where phosphate-buffered saline was used alone. These results may offer promising results for treating psoriasis with fingolimod, but we still need proper data on implementing this therapeutic in patients dealing with psoriasis [25].

### 3.2. The Characteristic of Sphingosine and Sphinganine (SFA)

In our study, we made an intriguing observation of significantly elevated concentrations of both sphinganine and sphingosine in psoriatic tissue in contrast with non-lesional and healthy skin samples. Notably, what set our findings apart was the discovery that, even in the non-affected skin of psoriatic patients, the concentration of sphingosine remained significantly higher than that observed in healthy skin. This highlighted a previously undescribed aspect of the lipid abnormalities present in psoriatic patients, underscoring the potential systemic involvement of sphingolipids in the disease.

During the de novo synthesis of ceramide, sphinganine is produced via the enzymatic conjunction of serine and palmitoyl-CoA by an enzyme called serine palmityltransferase. Sphinganine is then acylated to form ceramide. Ceramide is subsequently metabolized into glucosylceramide or sphingomyelin and can be further transformed into sphingosine and fatty acids through the action of ceramidase [17]. This enzymatic pathway plays a crucial role in the metabolism and regulation of sphingolipids in the body. Despite their relatively low stratum corneum levels (5–6% of total lipids), these sphingoid bases (SBs) (sphingosine and sphinganine) take part in the regulation of cell proliferation and differentiation, antimicrobial protection, and the upkeep of the integrity of lipid lamellae.

Previous studies investigating the composition of sphingosine and sphinganine in psoriatic skin are limited. However, our findings aligned with the observations made by Sung-Hyuk et al. [17], who reported elevated levels of sphingosine and sphinganine in psoriatic skin. Additionally, Sorokin et al. [26] reported similar results regarding sphinganine levels in psoriatic skin. These consistent findings across different studies provide further support for the involvement of sphingosine and sphinganine in the pathogenesis of psoriasis and highlight their potential significance as indicators or therapeutic targets in the disease [26].

Data regarding the sphingolipid composition of the skin in atopic dermatitis have been reported. Toncic et al. [27] found that levels of sphingosine and sphinganine were elevated in atopic dermatitis in comparison with healthy skin, although no similar differences were observed in non-lesional skin. Interestingly, our study revealed a similar elevation of sphingosine and sphinganine in psoriatic lesions, but we additionally observed an increase in sphingosine levels in non-lesional psoriatic skin. In atopic dermatitis, the ratio of sphinganine over sphingosine presented a progressive reduction from healthy skin to non-lesional and lesional atopic dermatitis skin, as reported by Toncic et al. [27]. These findings highlighted the distinct sphingolipid alterations in different skin conditions and further emphasized the complex role of sphingolipids in atopic dermatitis and psoriasis.

Indeed, Toncic et al. [27] spotted a depletion in the ratio of sphinganine over sphingosine in both lesional and non-lesional skin of individuals with atopic dermatitis. This reduction in the ratio was found to be inversely correlated with the severity of atopic dermatitis in unaffected and lesional skin. These findings suggested that alterations in the sphinganine/sphingosine ratio might play a part in the pathogenesis and severity of atopic dermatitis, highlighting the potential importance of sphingolipid metabolism in this skin condition [27].

In an interesting study by Arikawa et al. [28], it was demonstrated that altered ceramide metabolism in atopic dermatitis may be associated with increased vulnerability to colonization by Staphylococcus aureus in atopic dermatitis patients. The researchers found that levels of sphingosine were significantly decreased in both uninvolved and involved stratum corneum of atopic dermatitis patients compared with healthy individuals. Importantly, this decrease in sphingosine levels was correlated with an increased presence of bacteria, including S. aureus, in the upper stratum corneum of the same atopic dermatitis patients. These findings suggested a potential link between impaired sphingosine metabolism, altered skin barrier function, and increased susceptibility to bacterial colonization in atopic dermatitis [28].

In clinical practice, it is generally observed that secondary skin infections are less common in psoriasis compared with atopic dermatitis. Based on our study, the increased levels of sphingosine in both non-affected and lesional skin samples may potentially be related to a lower risk of skin infection in psoriasis. These results indicated that sphingolipids, such as sphingosine, may play a role in maintaining the integrity of the skin barrier and providing some level of protection against infections in psoriasis. However, subsequent study is needed to fully describe the intricate association between sphingolipids, skin barrier function, and susceptibility to skin infections in psoriasis.

### 3.3. The Characteristic of Ceramide

In our study, we observed an elevation of ceramide in psoriatic skin in comparison with the skin of healthy individuals. Tawada et al. [29] informed that ceramide levels were up to the constant sustainability between their output and degradation. The manufacture of ceramide in psoriasis was impaired, presumably due to the reduced activity of ceramide synthase [29]. Alessandrini et al. [30] showed that, additionally, abnormal ceramide levels were due to decreased sphingomyelinase activity (another crucial enzyme engaged in ceramide synthesis) and a reduced amount of prosaposin (a saposin precursor), which was an essential factor in the process of hydrolysis of sphingolipids [30]. Referring to Lew et al. [15], in psoriasis, the rate of ceramide in the epidermis was reduced, along with protein kinase C alpha and c-Jun N-terminal kinase. The reduction of ceramides can cause the downregulation of both these kinases, which are apoptotic cell-signaling molecules. This may result in a reduction of sphingomyelinase-induced ceramide generation and an enhancement of the proliferative and anti-apoptotic characteristics of the psoriatic epidermis. [15].

In the study performed by Checa et al. [13], they similarly described an elevated amount of ceramide in the skin of psoriatic patients in contrast with the skin of healthy individuals. This finding was consistent with our study. However, in contrast, Cho et al. [31] found that the rate of ceramides in the lesional epidermis of psoriasis patients was more reduced than in the non-lesioned epidermis. These contradictory findings highlighted the complex nature of ceramide metabolism in psoriasis and the potential variations that can exist between different studies. It is important to consider that various factors, including disease severity, sample collection methods, and patient characteristics, may contribute to these discrepancies [31]. In comparison with our study, Cho et al. [31] examined a smaller group of patients than ours (10 patients vs. 17 patients), which could lead to the conclusion that our study was more precise compared with that of Cho et al., because we examined a larger number of patients suffering from psoriasis. Nevertheless, studies on larger groups of patients should be performed. Toncic et al. [27] also observed elevated levels of ceramide and glucosylceramide in atopic dermatitis skin. These alterations were more prominent in psoriatic lesions, but significantly varied levels were also spotted in non-lesional skin. Interestingly, in our study on psoriatic patients, we made similar observations where the ceramide concentration was higher in the lesioned skin compared with non-lesioned skin samples and healthy skin. These results indicated that alterations in ceramide metabolism may be a common feature in inflammatory skin diseases, for instance, atopic dermatitis and psoriasis. The dysregulation of ceramide levels could contribute to the compromised skin barrier function and inflammatory processes observed in these diseases. Further research is needed to unravel the specific mechanisms underlying ceramide’ metabolism in different skin disorders and its implications for disease pathogenesis and potential therapeutic interventions.

In summary, there were similarities between the findings of our study conducted on psoriatic patients and the study conducted on atopic dermatitis patients regarding the differences in the levels of SBs and their ceramide among healthy and affected skin. In both conditions, the changes in SBs and ceramides were more prominent in the lesional skin and were associated with the severity of the disease, indicating the involvement of the immune and inflammatory responses in ceramide metabolism. Interestingly, in non-lesional skin, changes in ceramide composition were also observed, suggesting that alterations in ceramide metabolism may exist even in healthy skin. These findings highlighted the importance of studying ceramide metabolism and its clinical implications [27].

In psoriatic patients, elevated levels of ceramide were observed in the serum. However, the relationship between ceramides in the skin tissue and their influence on serum ceramide levels is not yet fully understood. It remains unclear how alterations in skin ceramide metabolism contribute to changes in serum ceramide levels in psoriasis. Ceramides have been implicated in various comorbidities, including components of metabolic syndrome. Hao et al. [32] informed that metabolic syndrome was a group of abnormalities that enhanced the chance of cardiovascular disease and diabetes. Lee et al. [33] indicated that ceramides have been associated with insulin resistance, dyslipidemia, obesity, and other metabolic abnormalities [33]. However, the specific mechanisms linking ceramides to these comorbidities and their role in psoriasis-related metabolic disturbances require further investigation.

There were several limitations to our study. Firstly, we focused on the measurement of total ceramide levels without assessing the specific subtypes or compositions of ceramides. Different ceramide subtypes may have distinct roles and effects in psoriasis and their contributions to the disease process remain to be elucidated. Future studies could explore specific ceramide subtypes and their associations with psoriasis severity and clinical outcomes. Secondly, the sample size in our study was relatively small. While our findings provided important insights into sphingolipid alterations in psoriasis, larger studies involving a bigger quantity of patients are necessary to establish the significance and severity of sphingolipids in the context of psoriasis. A larger sample size would enhance the statistical power and allow for more robust conclusions. Additionally, our study focused on a specific cohort of patients; thus, the generalizability of our findings to other populations or ethnicities may be limited. Future studies should aim to include diverse patient populations to ensure the broader applicability of the results.

## 4. Materials and Methods

A total of 15 patients (7 males and 8 females) with active plaque-type psoriasis, at a median age of 51.0 (43.0–66.0) and 17 healthy controls (11 males and 6 females) at a median age of 42.0 (35.5–55.0) were enrolled in the study. The severity of psoriasis was estimated using psoriasis area and severity index (PASI) [34]. Body mass index (BMI) was calculated based on self-reported weight and height. None of the patients or controls were under dietary restriction. History of hypertension, liver disease (e.g., non-alcoholic fatty liver disease (NAFLD)), heart disease, diabetes mellitus, and results of the laboratory tests were collected from hospital records of the patients. Laboratory tests were measured before the treatment. All psoriatic patients gave their written informed consent before enrolment in the study. The study protocol was approved by the local university bioethical committee (no. APK.002.500.2021) according to the principles of the Helsinki Declaration. Peripheral blood samples were taken before starting the treatment after overnight fasting. After centrifugation, the serum was stored at −80 °C until it was analyzed. A size of 3 mm punch biopsies were obtained from both the non-lesional and lesional skin from the trunk of psoriatic patients following local anesthesia. Samples from healthy patients were collected from the wound edge during a planned inguinal hernia operation under general anesthesia. Furthermore, in line with the PASI score, we assessed the clinical features of erythema, induration, and desquamation within the chosen psoriatic lesion for biopsy. The scoring system ranged from 0 to 4.

### 4.1. Sphingolipid Analysis

The content of sphingosine, sphinganine, sphingosine-1-phosphate, and SA1P was measured simultaneously according to the method of Min et al. (2002) [35]. Briefly, tissues were homogenized in a solution composed of 25 mM HCl and 1 mM NaCl. Acidified methanol and internal standards (10 pmol of C17-sphingosine and 30 pmol of C17-sphingosine-1-phosphate; Avanti Polar Lipids (Alabaster, AL, USA)) were added and the samples were ultrasonicated in ice-cold water for 1 min. Lipids were then extracted by the addition of chloroform, 1 M NaCl and 3 N NaOH. The alkaline aqueous phase containing sphingosine-1-phosphate and SA1P was transferred to a fresh tube. The residual phosphorylated sphingoid bases in the chloroform phase were reextracted twice with methanol/1 M NaCl (1:1, *v*/*v*) solution and then all the aqueous fractions were combined. The amount of sphingosine-1-phosphate and SA1P was determined indirectly after dephosphorylation to sphingosine and sphinganine, respectively, with the use of alkaline phosphatase (bovine intestinal mucosa; Fluka (Seelze, Germany)). To improve the extraction yield of released sphingosine and sphinganine, some chloroform was carefully placed at the bottom of the reaction tubes. The chloroform fractions containing free sphingosine and sphinganine or dephosphorylated sphingoid bases were washed three times with alkaline water (pH adjusted to 10.0 with ammonium hydroxide) and then evaporated under a nitrogen stream. The dried lipid residues were redissolved in ethanol, converted to their o-phthalaldehyde derivatives, and analyzed using a high-performance liquid chromatography (HPLC) system (PROSTAR; Varian Inc. (Palo Alto, CA, USA)) equipped with a fluorescence detector and C18 reversed-phase column (OMNISPHER 5, 4.6·150 mm; Varian Inc.). The isocratic eluent composition of acetonitrile: water (9:1, *v*/*v*) (Merck, Darmstadt, Germany) and a flow rate of 1 mL min^−1^ were used.

The content of ceramide was determined by the procedure previously described by Baranowski et al. [36]. A small volume of the chloroform phase, containing lipids extracted as described above, was transferred to a fresh tube containing 31 pmol of C17-sphingosine as an internal standard. The samples were evaporated under a nitrogen stream, redissolved in 1 M KOH in 90% methanol, and heated at 90 °C for 60 min to convert ceramide into sphingosine. This digestion procedure does not convert complex sphingolipids, such as sphingomyelin, galactosylceramide, or glucosylceramide, into free sphingoid bases (Bose et al., 1998). Samples were then partitioned by the addition of chloroform and water. The upper phase was discarded and the lower phase was evaporated under nitrogen. The content of free sphingosine liberated from ceramide was then analyzed using HPLC, as described above. The calibration curve was prepared using N-palmitoylsphingosine (Avanti Polar Lipids) as a standard. The chloroform extract used for the analysis of ceramide levels also contained small amounts of free sphingoid bases. Therefore, the content of ceramide was corrected for the level of free sphingosine determined in the same sample.

### 4.2. Statistical Analysis

The obtained data were analyzed using R (ver. 4.2.2) statistical package. At the onset of analyses, the continuous data were checked for normality (Shapiro–Wilk test) and homoscedasticity (Fligner–Killeen test). Based on the above, the between-group comparisons were made using parametric (ANOVA with subsequent pairwise Student’s *t*-tests) or non-parametric methods (Kruskall–Wallis test with subsequent pairwise Wilcoxon tests). The obtained *p*-values were adjusted for multiple comparisons (Benjamini–Hochberg correction). Corrected *p*-values lower than 0.05 were considered to be statistically significant. The results were presented in the form of box–whisker plots. The dependence between the variables of interest was determined based on Pearson’s coefficients and depicted using heatmaps.

## 5. Conclusions

The observed differences in the profile of bioactive sphingolipids between psoriatic skin and healthy skin indicated a significant contribution of these lipids in the development of inflammation in psoriasis. These differences, which were more pronounced in psoriatic skin, highlighted the potential influence of the studied parameters on the inflammatory processes associated with the disease.

Furthermore, the finding that sphingolipid metabolism was impaired not only in the affected skin but also in the clinically unchanged skin of psoriasis patients suggested a broader systemic involvement of lipid dysregulation in psoriasis. This implied that lipid metabolism abnormalities may extend beyond the visible psoriatic lesions and might involve the overall pathogenesis of the disease. Subsequent research is needed to fully understand the consequences of these lipid abnormalities and their potential as therapeutic targets in psoriasis management.

## Figures and Tables

**Figure 1 ijms-24-11336-f001:**
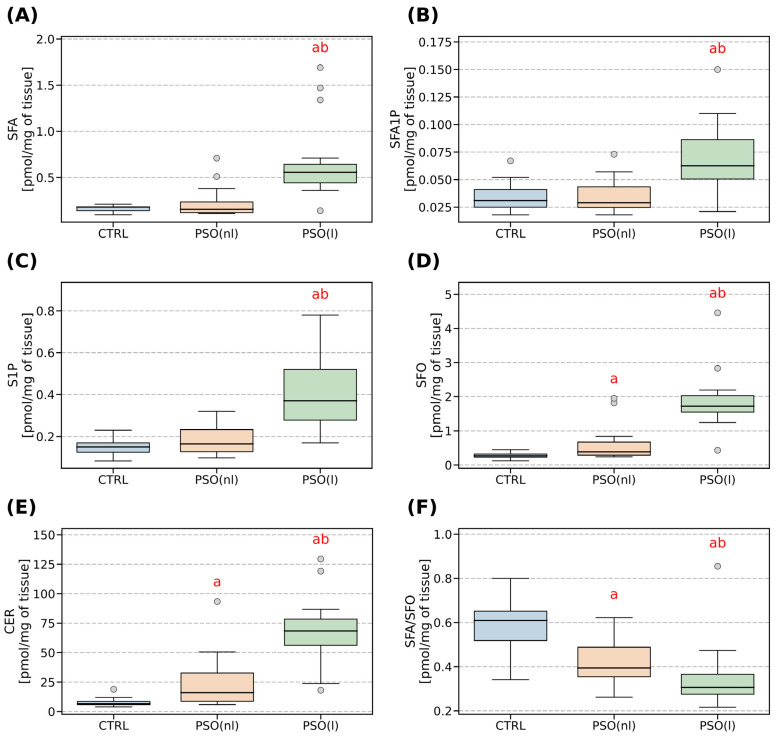
Comparison between sphingolipids in healthy (CTRL), psoriatic lesional (PSO (l)), and psoriatic non-lesional (PSO (nl)) skin. (**A**) Amount of sphinganine (SFA) (pmol/mg of tissue). (**B**) Amount of sphinganine-1-phosphate (SFA1P) (pmol/mg of tissue). (**C**) Amount of sphingosine-1-phosphate (S1P) (pmol/mg of tissue). (**D**) Amount of sphingosine (SFO) (pmol/mg of tissue). (**E**) Amount of ceramide (CER) (pmol/mg of tissue). (**F**) The ratio of sphinganine and sphingosine. Significance markers: a signifies different vs. CTRL (*p* < 0.05) and b signifies different vs. PSO (NL) (*p* < 0.05).

**Figure 2 ijms-24-11336-f002:**
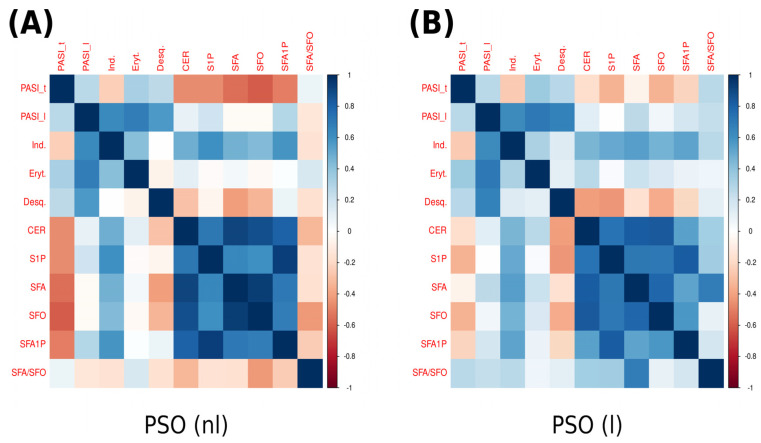
Correlation matrix (heatmap). Pearson correlation coefficients are depicted as shades of blue (positive correlation) or red (negative correlation). (**A**) Correlation matrix for psoriatic non-lesional (PSO (nl)) skin. (**B**) Correlation matrix for psoriatic lesional (PSO (l)) skin. CER—ceramide; Desq.—desquamation; Eryth.—erythema; Ind.—induration; PASI_l—psoriasis area severity index_lesional; PASI_t—psoriasis area severity index_total; S1P—sphingosine-1-phosphate; SFA—sphinganine; SFA1P—sphinganine-1-phosphate; SFO—sphingosine.

**Table 1 ijms-24-11336-t001:** Clinical and biochemical characteristics of the control group (CTRL) and psoriatic patients (PSO). Data are presented as median and interquartile range; a signifies different vs. PSO (*p* < 0.05); BMI—body mass index, CRP—C reactive protein, TAG—triacylglycerol, AST—aspartate transaminase, ALT—alanine transaminase.

Clinical and Laboratory Features	CTRL (*n* = 17)	PSO (*n* = 15)
**Age (years)**	42.0 (35.5–55.0)	51.0 (43.0–66.0)
**Body mass (kg)**	75.0 (67.5–78.0)	87.0 (82.0–94.0) a
**Height (cm)**	174.0 (166.5–176.0)	174.0 (162.0–176.0)
**BMI (kg/m^2^)**	25.06 (23.75–27.31)	28.74 (27.72–30.35) a
**CRP (mg/dL)**	1.0 (1.0–1.45)	4.65 (3.06–8.11) a
**Glucose (mg/dL)**	93.0 (88.5–100.0)	84.0 (81.0–94.0)
**TAG (mg/dL)**	73.0 (67.5–82.0)	122.0 (85.0–135.0) a
**AST (U/L)**	22.0 (17.5–28.5)	20.0 (19.0–32.0)
**ALT (U/L)**	17.0 (13.0–22.0)	18.0 (15.0–27.0)
**Sex (no. female/no. male)**	6/11	8/7

## Data Availability

The data presented in this study are available on request.

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
