# Peer review of "The Interplay between Bioactive Sphingolipids in the Psoriatic Skin and the Severity of the Disease"

_ijms, 2023, doi:10.3390/ijms241411336_

Round 1

Reviewer 1 Report

In this article, the authors investigated the levels of various sphingolipids in psoriatic patients and reported increased levels of sphingolipids in psoriatic skin compared with non-lesional skin of psoriatic patients as well as healthy controls. Although the data seem intriguing, the manuscript needs revision to be published in the journal.

[specific comments]

#1. The numbers of psoriatic patients and healthy controls are described to be sixteen and sixteen in Abstract, but are described fifteen and seventeen in Materials and Methods. It needs to be corrected.

#2. In Materials and Methods, the method by which samples of non-lesional areas were collected from psoriatic patients seems to be missing.

#3. In Results #3.1, the percentages of overweight, obesity, normal weight, and different PASI scores seem to be incorrect.

#4. In Results #3.2 (lines 214-215), although it is described that “SFA, S1P, and SFA1P were higher in both lesional and non-lesional skin”, Figure 1 indicates that in non-lesional skin SFO was higher compared to healthy control, but SFA, S1P, and SFA1P were not.

#5. In Results #3.2, the description on the relationship between PASI scores and sphingolipids seems to be missing.

#6. In Discussion (line 384), “Moon” seems to be the first name.

#7. In Discussion (line 391, 396, 401), the reference number for Toncic is 34, not 30.

#8. In Discussion (line 436), “prosaposin” is a precursor of “saposin”, not “saponin”.

Moderate editing of English language should be required.

Author Response

In this article, the authors investigated the levels of various sphingolipids in psoriatic patients and reported increased levels of sphingolipids in psoriatic skin compared with non-lesional skin of psoriatic patients as well as healthy controls. Although the data seem intriguing, the manuscript needs revision to be published in the journal.

Dear Reviewer,

Thank you for the time spent on the revision of our article and for the constructive remarks. I have made the corrections that you have suggested and hope that with your help we managed to improve our manuscript.

Best regards,

Mateusz Matwiejuk

[specific comments]

#1. The numbers of psoriatic patients and healthy controls are described to be sixteen and sixteen in Abstract, but are described fifteen and seventeen in Materials and Methods. It needs to be corrected.

Thank you very much for your remark. According to your suggestion, I have corrected it.

#2. In Materials and Methods, the method by which samples of non-lesional areas were collected from psoriatic patients seems to be missing.

Thank you very much for your remark. According to your suggestion, I have corrected it. I clarified it the Materials and Methods. 

#3. In Results #3.1, the percentages of overweight, obesity, normal weight, and different PASI scores seem to be incorrect.

Thank you very much for your remark. According to your recommendation, I have corrected it. I put my remarks in the table, in the Materials and Methods. 

#4. In Results #3.2 (lines 214-215), although it is described that “SFA, S1P, and SFA1P were higher in both lesional and non-lesional skin”, Figure 1 indicates that in non-lesional skin SFO was higher compared to healthy control, but SFA, S1P, and SFA1P were not.

Thank you very much for raising this issue. According to your suggestion, I have corrected it. I have corrected in the manuscript, in the Results section.

#5. In Results #3.2, the description on the relationship between PASI scores and sphingolipids seems to be missing.

Thank you very much for your suggestion. According to your remark, I have added this information in the Results section. This information was previously written in the Discussion.

#6. In Discussion (line 384), “Moon” seems to be the first name.

Thank you very much for your remark. You are right Moon is the first name, and I have changed into surname, Sung-Hyuk.

#7. In Discussion (line 391, 396, 401), the reference number for Toncic is 34, not 30.

Thank you very much for your remark. Based on your suggestion, I have revised the numbers to 27, as we needed to reorganize our manuscript to comply with the MDPI policy, which involved moving the Materials and Methods chapter to the end of the manuscript.

#8. In Discussion (line 436), “prosaposin” is a precursor of “saposin”, not “saponin”.

Thank you very much for your suggestion. According to your remark, I have changed it from saponin to saposin.

Reviewer 2 Report

Matwiejuk nicely show the correlation between sphingolipid levels and psoriasis and how this  correlation relates to the severity of the disease. These findings are relevant and can be referred into future for detection, and even maybe for the prevention of the disease. The manuscript is suitable for publication, I only have minor, textual comments:

(1) The difference between ceramides and ceramide. The authors use the word “ceramides” in the manuscript, which is plural of “ceramide”, unless they refer to multiple ceramide species. The sentence in the abstract “…ceramides, sphingosine-1-phosphate, sphingosine, sphingomyelin…” should be “ceramide”. This should be fixed, including the keywords.

Similar pattern follows the manuscript, e.g.

Line 77: “analyzed ceramides C16:0, C18:0, C20:0, C22:0, and C24:1” is correct.

Line 82: “levels of sphingomyelins were altered” should be “sphingomyelin”.

Line 84: C16:0-, C24:1-, and C24:0-sphingomyelins” is also correct.

(2) Decimals should be separated by period not by comma, e.g.

Line 2: 0,27% to 11,4% should be “0.27% to 11.4% and so.

(3) The text on the figures could be improved, they are not well readable.

(4) The manuscript contains too many unnecessary abbreviations. They are useful in the figures and legends but the main text does not need this many abbreviations, e.g. CER, SFO, SFA1P, PASI_t, S1P… are not needed. MS and AD are used only a few times in the text.

(5) Typo in the abstract sphingosine-1-, phosphate” should be “sphingosine-1-phosphate”.

Author Response

Matwiejuk nicely show the correlation between sphingolipid levels and psoriasis and how this  correlation relates to the severity of the disease. These findings are relevant and can be referred into future for detection, and even maybe for the prevention of the disease. The manuscript is suitable for publication, I only have minor, textual comments:

Dear Reviewer,

Thank you very much for the revision of our article and for your constructive comments. I have made the corrections according to your suggestions and I hope that with your help we were able to improve our manuscript.

Best regards,

Mateusz Matwiejuk

(1) The difference between “ceramides” and “ceramide”. The authors use the word “ceramides” in the manuscript, which is plural of “ceramide”, unless they refer to multiple ceramide species. The sentence in the abstract “…ceramides, sphingosine-1-phosphate, sphingosine, sphingomyelin…” should be “ceramide”. This should be fixed, including the keywords.

Similar pattern follows the manuscript, e.g.

Line 77: “analyzed ceramides C16:0, C18:0, C20:0, C22:0, and C24:1” is correct.

Line 82: “levels of sphingomyelins were altered” should be “sphingomyelin”.

Line 84: “C16:0-, C24:1-, and C24:0-sphingomyelins” is also correct.

 Thank you very much for your remarks. According to your suggestions, I have carefully corrected it according to your recommendations.

(2) Decimals should be separated by period not by comma, e.g.

Line 2: “0,27% to 11,4%” should be “0.27% to 11.4%” and so.

Thank you very much for your suggestion. I have made improvements to the manuscript based on your recommendations.

(3) The text on the figures could be improved, they are not well readable.

Thank you very much for your remark. Following your recommendation, I have included a newer figure in the manuscript with improved visibility and a larger font size.

 (4) The manuscript contains too many unnecessary abbreviations. They are useful in the figures and legends but the main text does not need this many abbreviations, e.g. CER, SFO, SFA1P, PASI_t, S1P… are not needed. MS and AD are used only a few times in the text.

  Thank you very much for your suggestion. Following your remark, I have eliminated those abbreviations and replaced them with their corresponding full names.

(5) Typo in the abstract “sphingosine-1-, phosphate” should be “sphingosine-1-phosphate”.

 Thank you very much for your issue. According to your remark, I have corrected it from “sphingosine-1-, phosphate” to “sphingosine-1-phosphate”.

Round 2

Reviewer 1 Report

In the revised version of the manuscript, the authors have corrected all points I commented on. However, I noticed other mistakes in the manuscript. Sphingosine-1-"phosphatase" should be sphingosine-1-"phosphate". Similarly, ceramide-1-"phosphatase" should be ceramide-1-"phosphate". Please correct them.

Author Response

In the revised version of the manuscript, the authors have corrected all points I commented on. However, I noticed other mistakes in the manuscript. Sphingosine-1-"phosphatase" should be sphingosine-1-"phosphate". Similarly, ceramide-1-"phosphatase" should be ceramide-1-"phosphate". Please correct them.

Once more, I would like to thank you very much for your remark. According to your suggestion, I have corrected it.